# Design and Validation of RT-PCR Assays to Differentiate *Salmonella* Vaccine Strains from Wild-Type Field Isolates

**DOI:** 10.3390/vetsci11030120

**Published:** 2024-03-06

**Authors:** Pieter-Jan Ceyssens, Doris Mueller-Doblies, Wesley Mattheus

**Affiliations:** 1Unit of Human Bacterial Diseases, Sciensano, 1050 Brussels, Belgium; pieter-jan.ceyssens@sciensano.be; 2Elanco Austria GmbH, Quartier Belvedere Central, 1100 Vienna, Austria; doris.mueller_doblies@elancoah.com

**Keywords:** salmonella, vaccination, poultry

## Abstract

**Simple Summary:**

The timely differentiation of the AviPro Salmonella VAC T and VAC E strains from the wild-type *Salmonella enterica* ser. Typhimurium and ser. Enteritidis isolates is crucial for effectively monitoring veterinary isolates from poultry. In this study, we developed two triplex Real-Time PCR reactions that targeted conserved and specific mutations and, therefore, enabled the reliable differentiation of field and vaccine strains. This validated method demonstrated a 100% sensitivity and specificity for distinguishing both *Salmonella enterica* ser. Typhimurium and Enteritidis and can serve as an alternative method for those laboratories that prefer a PCR-based method over the phenotypic antimicrobial resistance testing that is currently used.

**Abstract:**

The timely differentiation of the AviPro Salmonella VAC T and VAC E strains from the wild-type *Salmonella enterica* ser. Typhimurium and ser. Enteritidis isolates is crucial for effectively monitoring veterinary isolates. Currently, the distinction between field and vaccine strains has been conducted routinely via phenotypic antimicrobial resistance testing since the vaccines were first introduced more than 20 years ago, and the differentiation based on the antimicrobial resistance profile is still a valid and well-established method. However, an alternative method was sought for those laboratories that prefer a PCR-based method for logistic and/or operational reasons. In this study, we developed two triplex Real-Time PCR reactions that targeted conserved and specific mutations and, therefore, enabled the reliable differentiation of field and vaccine strains. To validate the effectiveness of both assays, we extensively tested them on a dataset consisting of 405 bacterial strains. The results demonstrated a 100% sensitivity and specificity for distinguishing both *Salmonella enterica* ser. Typhimurium and Enteritidis, although a confirmed culture is required.

## 1. Introduction

The *Salmonella enterica* serovars Typhimurium (*Salmonella* ser. Typhimurium) and Enteritidis continue to pose a significant global threat as one of the leading causes of foodborne illnesses. These bacterial species are commonly contracted by consuming contaminated food of animal origin, such as poultry, eggs, and meat [1,2]. To address this public health concern, the European Union has implemented regulations and set targets for the reduction in the serovars Enteritidis and Typhimurium in various poultry populations, including breeding hens, laying hens, broilers, breeding turkeys, and fattening turkeys [3].

Achieving successful control of *Salmonella* requires a comprehensive approach that encompasses several key elements. First and foremost, implementing good farming practices and maintaining high standards of hygiene are crucial for preventing and minimizing the spread of the bacterium within poultry flocks. Regular testing is also essential to detect any potential *Salmonella* contamination, and the process of eliminating positive flocks can be costly and time-consuming [4]. In addition to these measures, the use of vaccines against *Salmonella* spp. is considered an important strategy to enhance the resistance of poultry to *Salmonella* exposure and reduce bacterial shedding. Vaccination provides an additional layer of protection and can contribute to achieving the set targets for *Salmonella* reduction [5]. Currently, EU legislation mandates the vaccination of laying hens against the serovar Enteritidis if the national prevalence exceeds 10% based on monitoring programs. Other vaccinations against *Salmonella* are voluntary but can play a significant role in supporting the overall control efforts. Most European countries allow and support the use of live vaccines in poultry, and some even offer subsidies [6].

The attenuationed drift mutant strains Avipro^TM^ Salmonella VacE (providing homologous protection against the serovar Enteritidis) and Avipro^TM^ Salmonella VacT (providing homologous protection against the serovar Typhimurium) have been successfully used to protect breeders and layers since the 1990s. Since 2012, a combination product, Avipro^TM^ Salmonella DUO, containing both strains has been registered and offers the convenience of homologous protection against both serovars [7]. To date, the use of these live attenuated vaccines has been preferred because they induce a broader host immune response compared to inactivated vaccines and provide protective immunity through both cell-mediated and humoral immune responses [8].

In order to be safely used in the field, a reliable method is necessary for every live vaccine to be distinguished from field strains. If this was not possible, the detection of the live *Salmonella* serovar Enteritidis or Typhimurium vaccine strains in samples from poultry would trigger undesired consequences, such as the withdrawal of the product from the market or even culling of the affected flock. Furthermore, in the European Union, regulation 1177/2006 [3] requires that live vaccines for poultry be distinguished from field strains.

So far, differentiation of the two vaccine strains has been achieved using their antimicrobial resistance pattern, which is still a valid and widely used method. Both strains show sensitivity to erythromycin, while field *Salmonella* strains are intrinsically resistant to erythromycin, and both show resistance to rifampicin. The Avipro^TM^ Salmonella VacE strain has an additional resistance to streptomycin, while the Avipro^TM^ Salmonella VacT strain is resistant to nalidixic acid, which helps with differentiating between the two strains [9]. Even though the differentiation between field and vaccine strains based on the antimicrobial resistance pattern is a well-established method, there was the perceived demand from customers to have a PCR-based differentiation methodology available, as PCR, and in particular Real-Time PCR, is commonly used nowadays in most laboratories. PCR is seen as an easy-to-use method that does not require specialist technical expertise and can be interpreted using a straight-forward read-out matrix [10,11].

The limitation of culture-based distinction has prompted previous development of a Luminex-based assay to distinguish the vaccine from wild-type strains [12]. However, this assay was limited to the serovar Typhimurium and lacked universal applicability given the requirement of a specific MagPix™ platform. In this study, we designed and validated a Real-Time PCR assay that is able to discriminate field and vaccine strains of both the serovars Typhimurium and Enteritidis with 100% accuracy, starting from an overnight culture of confirmed *Salmonella* ser. Typhimurium/Enteritidis strains.

## 2. Materials and Methods

### 2.1. Bacterial Strains

All bacterial strains used in this study are listed in Appendix A. *Vibrio alginolyticus* M/5035 was used as negative control strain. Bacterial cultures were grown overnight at 37 °C on nutrient agar (Bio-Rad Laboratories, Hercules, CA, USA). Bacterial species identity was confirmed using MALDI-TOF (Bruker Biotyper, IVD Database 5.0, Bruker, Kontich, Belgium). *Salmonella* serotypes were confirmed using slide agglutination using commercial antisera, and interpreted according to the White–Kauffmann–Le Minor scheme [13]. For DNA extraction, a single colony was added to 50 µL of DI H_2_O and placed in a thermal cycler (90 °C for 10 min, cool to 4 °C). The mixture was spun (13,000× *g*, 10 min), and the supernatant was used immediately or stored at −20 °C. During validation, the DNA extract was diluted 1:100 (*v*/*v*) using nuclease free water.

Bacterial strains analyzed for their specificity and sensitivity included 60 *Salmonella* ser. Enteritidis strains and 120 *Salmonella* ser. Typhimurium strains (60 monophasic variants and 60 biphasic variants), which were randomly selected from National Reference Laboratory collected, and serotyped according to a ISO 17025 validated White–Kauffmann–Le Minor procedure [14]; these serovars, taken together, represent around three quarters of human *Salmonella* isolates in Europe [15]. Furthermore, between five and ten isolates of each of the ten most common serovars found in poultry samples and human specimens in Europe were included (Appendix A). The isolates were chosen from different sources/different outbreaks to ensure that a wide genetic variety of isolates was represented.

### 2.2. RT-PCR

For all targeted genes, primers and probes were designed using Visual OMP™ software (version 7.6.58.0; DNA Software, Plymouth, MA, USA) against the reference strains of serovars Typhimurium (strain LT2, NC_003197.2) and Enteritidis (strain P125109, NC_011294.1) and ordered from IDT Technologies (PrimeTime probes, Heverlee, Belgium) and Thermo Scientific (TaqMan QSY, Dilbeek, Belgium). In the two multiplexes (VacE/VacT), the final primer and probe concentrations were 0.5 and 0.25 µM, respectively, in a total reaction volume of 20µL using ready-to-use Taqman Mastermix (BactoPure, Thermo Scientific, Waltham, MA, USA). After initial heating step (10 min at 95 °C), we ran 40 cycles of denaturation (15 s at 95 °C) and annealing/extension (60 s at 60 °C) on a Quantstudio 5 (Thermo Scientific, USA), with Ct values < 36 being considered positive. The test sensitivity and specificity was calculated using strains previously confirmed as belonging to either the serovar Enteritidis or Typhimurium (True and False Positive (TP/FP) and True and False Negative (TN/FN)) and the vaccine strains from the supplier. The test Accuracy was 100% × (TP + TN)/(TN + FN + TP + FP) and described the likelihood that the results of the assay were correct. Test sensitivity (100% × TP/(TP + FN)) described the likelihood that a result would be correctly picked up by the assay when present. Finally, test specificity (100% × TN/(TN + FP)) described the likelihood that a result would not be falsely picked up by the assay when not present.

## 3. Results and Discussion

### 3.1. Assay Development

We designed two triplex RT-PCR reactions that were able to reliably distinguish field samples from AviPRO vaccine strains (Table 1, Figure 1). The targeted genetic regions that served to confirm the presence of *Salmonella* spp. and either the ser. Typhimurium or ser. Enteritidis were derived from previous works [16,17].

The selection of a specific genetic markers for the AviPro SALMONELLA VAC T strain was based on its reduced erythromycin susceptibility. This deletion results in a premature termination codon (Leu876fs) in the gene encoding the AcrB subunit of the RND transporter permease, leading to compromised drug efflux. Additionally, within the same gene, another deletion mutation (acrB:912_916del) was identified. On the other hand, for distinguishing field strains from the AviPro SALMONELLA VAC E strain, other researchers demonstrated that single-nucleotide polymorphisms (SNPs) in *kdpA* proved to be the most effective differentiating factors between the two [10].

### 3.2. Assay Validation

Next, we evaluated the sensitivity (using inclusivity tests), the specificity (using exclusivity tests), and the accuracy of both assays. Apart from both vaccine strains, we used a large selection of wild-type strains of the *Salmonella* ser. Enteritidis (*n* = 60) and the *Salmonella* ser. Typhimurium (*n* = 60) and its monophasic variant (*n* = 60). Human isolates were selected for maximal diversity in MLVA profiles, while veterinary strains were sampled at random from national monitoring programs for pigs and poultry. To test exclusivity, we included 200 non-target *Salmonella* serovars and 25 non-*Salmonella* strains in the panel. All results, including the Ct values, are summarized in Appendix A.

This analysis showed that the dedicated assays correctly identified the AviPro SALMONELLA VAC T and VAC E strains (100% inclusivity) and excluded all the wild-type *Salmonella* ser. Typhimurium/Enteritidis strains (100% exclusivity). Additionally, both panels were robust, showing 100% reproducibility when retesting 12 random samples of the test panel by another laboratory technicians on different days. Likewise, both assays showed 100% repeatability when repeating 12 samples five times on independent occasions.

A more detailed analysis of the data revealed that the targeted *kdpA* point mutation reliably distinguished all the tested *Salmonella* field serovars (424/424) from the AVIPRO Vac E vaccine strain (Appendix A). The erythromycin deletion was also retrieved in all the Typhimurium and monophasic Typhimurium strains, as well as in isolates from the serovars Heidelberg (5/5), Indiana (1/5), Infantis (2/10) and Virchow (7/10). Additionally, 5.6% of tested genetic lineages of Enteritidis field strains were negative for the *ent* signal, and 5.3% of tested Typhimurium field strains were negative for the *rpoB* signal. Therefore, for the assay to work correctly, it should be applied only on confirmed isolates of *Salmonella enterica* ser. Enteritidis and Typhimurium.

## 4. Conclusions

To summarize, we have successfully created a fast molecular test capable of differentiating between wild-type *Salmonella* ser. Typhimurium and Enteritidis field isolates and the AviPro SALMONELLA VAC T and VAC E vaccine strains. This highly accurate test, with a 100% success rate, is set to replace the current phenotypic vaccine identification method at the National Reference Center in Belgium.

## Figures and Tables

**Figure 1 vetsci-11-00120-f001:**
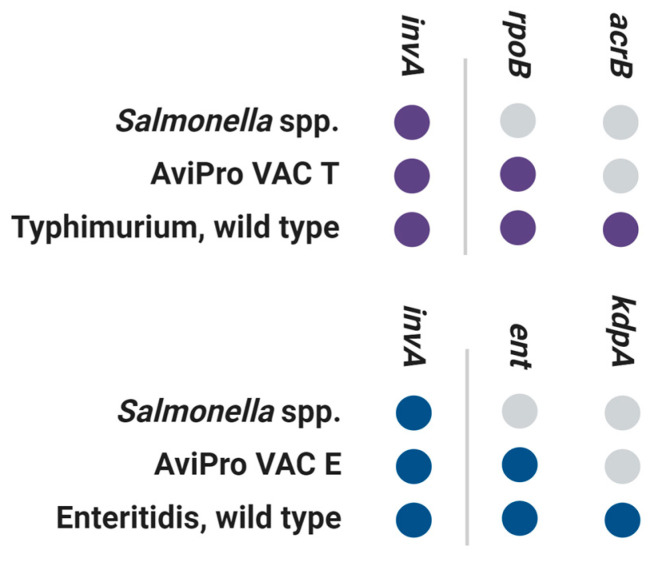
High-level design of the two triplex RT-PCR reactions for the discrimination of field and vaccine strains of the *Salmonella* ser. Typhimurium and Enteritidis. Colored dots indicate positive reactions.

**Table 1 vetsci-11-00120-t001:** Selected alleles and SNP positions for differentiating the AviPro Salmonella VAC E and VAC T strains from wild-type *Salmonella* ser. Typhimurium and ser. Enteritidis. A plus sign indicates a locked nucleotide.

Target	Name	Sequence	MIX	Rationale
*invA*	Probe	/56-FAM/ATCGCACCG/ZEN/TCAAAGGAACCGTAA/3IABkFQ/	E/T	Identify *Salmonella* spp.
	Forward	CACCGAAATACCGCCAATAAAG		
	Reverse	AGCGTACTGGAAAGGGAAAG		
*rpoB*	Forward	AACGGTACTGAGCGTGTTATC	T	Specific mutation for ser. Typhimurium
	Reverse	CTTTACCCGAAGAGTGGGTTT		
	Probe	/JUN/CACCGTAG+CCCTGGCGT/QSY7		
*acrB*	Forward	CATTACGGAGAACGGGATAGAC	T	Indel absent in VacT strain
	Reverse	CGTCAGGCATTGGGTATGA		
	Probe	/VIC/GCGATATAGCATACAGGG/QSY7		
*kdpA*	Forward	AGGACAAACAGCAACATG	E	Indel absent in VacE strain
	Reverse	GTATGGTGCCGATGTG		
	Probe	/VIC/CCAAAGACCACTTCGCCAA/QSY7		
*ent*	Forward	CCGCCCAGCTCATATTTCT	E	Specific mutation for ser. Enteritidis.
	Reverse	CGGATTCGAACCGACAGATT		
	Probe	/ABy/CCTCCGGCGGAAGTTCGTTAACAG/QSY7		

## Data Availability

Data are contained within the article and Appendix A.

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
