# Peer review of "Design and Validation of RT-PCR Assays to Differentiate Salmonella Vaccine Strains from Wild-Type Field Isolates"

_vetsci, 2024, doi:10.3390/vetsci11030120_

Round 1

Reviewer 1 Report

Comments and Suggestions for Authors

The authors in this work established two triplex-PCR to discriminate the salmonella from vaccine and field isolates. The authors are encouraged to address the comments below:

Application of this method. If this method is fully validated, is this method easier to be used in compared to the classical ones? How many labs will buy all these primers/probes from these two triplex PCR for this purpose?

The design of the right primers/probes are key for this work. The authors need to present the accession number of all selected sequences, and perform the alignment. Then, we can tell how you design the primers/probes....

The authors mentioned the 100% sensitivity for the established assays. I do not see how you can make this conclusion.

For multiplex PCR, there are always cross talks between the channels. Did you perform the color compensation to address the questions? In the real life, you have samples which contains field and vaccine Salmonella strain. Do you then detect both or one?

The title says the Salmonella vaccine. There are actually more Salmonella vaccine other than Avipro. The RT in RT-PCR means reverse-transcription. Did you mean to use this term?

Reviewer 2 Report

Comments and Suggestions for Authors

In this research Ceyssens et al., designed and validated two triplex RT-PCR assay which was able to discriminate field and vaccine strains of S. Typhimurium and S. Enteritidis with 100% accuracy. 

The authors conclude that this PCR can serve as an alternative method over the phenotypic antimicrobial resistance testing. 

The article has serious flaws and scientific lacks and must not be accepted for Veterinary Sciences. 

Some considerations are: 

- Salmonella nomenclature is different in each paragraph and it seems that the authors don't know this topic.  

Correct use in according to international nomenclature (examples): 

Salmonella subsp. enterica serovar Enteritidis 

Salmonella Typhimurium / S. Typhimurium

Enteritidis serovar / Typhimurium serotype

- There are some scientific terms that are not explained (i.e., exclusivity in M&M).

- Results and Discussion chapter should be named only 'Results'. There is a great lack of discussion with other author and researches. 

- Conclusions are too obvious (...)

- References are very very insufficient.

- In conclusion, this paper is based only in a specific RT-PCR, and in my opinion, has a poor scientific quality compatible with Vet. Sci. (MDPI).   

Comments on the Quality of English Language

The English is appropriate and grammatically correct except for some minor corrections. Authors must take care of some aspects of the text format and unify criteria, such as  adding a full stop in the headings of every section or not. 

Reviewer 3 Report

Comments and Suggestions for Authors

I review the paper presented by Pieter-Jan Ceyssens and colleagues titled ‘Design and validation of RT-PCR assays to differentiate Salmonella vaccine strains from wild-type field isolates’. To my evaluation, the paper should address the following to qualify for publication.

1/ Simpe summary and Abstract shouldn’t be a separate section. One abstract is required and enough.

2/ On the material and method section reference for the protocol or methodology is required. For example, bacterial culture. DNA extraction, according to which protocol or company?

3/ Primer information is required, and which region of the strain genome amplified and why? Primer design information is required and the company.

4/ Previous studies on sensitivity and specificity of the strain are required as a reference and for discussion? By what fold was the sensitivity and specificity of this test increased?

5/ Ct values and PCR amplification curves can be provided as a supplementary.  

Round 2

Reviewer 1 Report

Comments and Suggestions for Authors

This reviewer appreciates the detailed responses from the authors. I am happy with the corrections made by the authors. 

As the figures shared by the authors indicate, there are cross talks between these fluorescence channels. Each fluorescence channel has its peak with wide shoulders. When these shoulders touch each other's, we will see the spillover. Then, I am sure in the real life some chickens are infected with field isolates but also received vaccination. This is a big question which needs to be addressed. The authors need to consider mimicking this situation: mix different concentrations of vaccine DNA and field Salmonella DNA. Would you see something you expected?

Reviewer 2 Report

Comments and Suggestions for Authors

No additional comments
